Journal of
open psychology data

# Data from the Effects of Congruent and Incongruent Perceptual Cues on Middle Schoolers' Mathematical Performance, Learning, and Retention Study

DATA PAPER

ERIN OTTMAR ⓘ

PUYUAN ZHANG ⓘ

JI-EUN LEE ⓘ

JEFFREY K. BYE ⓘ

MAEGAN A. COLBERT ⓘ

ALENA EGOROVA ⓘ

SHUQI YU ⓘ

AVERY H. CLOSSER ⓘ

CAROLINE BYRD HORNBURG ⓘ

*Author affiliations can be found in the back matter of this article

]u[ ubiquity press

## ABSTRACT

This paper presents datasets for a research project that investigated the individual and combined effects of two perceptual cues—spacing and color—under varying conditions of congruence with the order of operations. The datasets contain 1,100 6th-grade students' data collected through a randomized controlled trial conducted in 2024 in a U.S. school district. All de-identified intervention data are openly available on the Open Science Framework. Additional datasets, including student demographics and assessment data, require a data sharing agreement. The data can be used by researchers interested in understanding perceptual learning mechanisms and improving online instructional materials for middle school mathematics.

**CORRESPONDING AUTHOR:**
**Erin Ottmar**

Department of Social Science and Policy Studies, Worcester Polytechnic Institute, US

erottmar@wpi.edu

**KEYWORDS:**
Perceptual cues; Mathematics learning; Middle school; Randomized controlled trial; Problem solving

**TO CITE THIS ARTICLE:**
Ottmar, E., Zhang, P., Lee, J.-E., Bye, J. K., Colbert, M. A., Egorova, A., Yu, S., Closser, A. H., & Hornburg, C. B. (2025). Data from the Effects of Congruent and Incongruent Perceptual Cues on Middle Schoolers' Mathematical Performance, Learning, and Retention Study. *Journal of Open Psychology Data,* 13: 5, pp. 1–16. DOI: https://doi.org/10.5334/jopd.139

# (1) BACKGROUND

## 1.1. BACKGROUND

Solving order-of-operations problems, which involve mathematical expressions with multiple operations (e.g., $4 - 7 \times 2 + 9$), is a crucial part of pre-algebra and algebra in middle school (Norton & Cooper, 2001), laying the groundwork for students' future math learning in high school. However, many middle school students struggle with these types of problems, particularly when the order-of-operations rules preclude solving the problem by performing calculations from left to right (e.g., $3 + 4 \times 2$; Bye et al., 2024). Building on Gibson's (1969) perceptual learning theory and Bjork's (1994) desirable difficulties, we developed an online intervention incorporating perceptual cues that highlighted the order of operations within worked examples and practice problems. The present study investigates the impact of this intervention on students' math performance and learning outcomes in regular classroom settings, as well as the role of student-level characteristics in the intervention's effects.

### 1.1.1. Perceptual Learning Theory

Perceptual learning theory (Gibson, 1969) posits that individuals naturally acquire knowledge by processing sensory information from their surroundings, such as visual and auditory cues. According to this theory, perception is an active and dynamic process in which individuals gradually improve their ability to identify key features through learning and repeated exposure. Perceptual learning plays a vital role in cognitively demanding tasks such as solving math problems, as students must effectively capture and process sensory cues, such as numbers and symbols, to arrive at accurate solutions (Closser et al., 2022; Goldstone et al., 2017; Landy & Goldstone, 2007).

Previous studies have documented the significance of interpreting perceptual cues in students' mathematical problem-solving (Closser et al., 2022; Landy & Goldstone, 2007; Ngo et al., 2023). For instance, manipulating the physical spacing between operational elements (e.g., $7 \times 6 + 4$) and using a different font color to highlight operational elements have been shown to influence students' mathematical performance (Alibali et al., 2018; Harrison et al., 2020). However, prior research has primarily focused on immediate outcomes, with limited investigation into the effects of perceptual cues on learning and long-term retention.

### 1.1.2. Desirable Difficulties

The use of congruent and incongruent perceptual cues also impacts the difficulty of operational problems. The concept of "desirable difficulty" (Bjork, 1994) describes learning conditions that impose a temporary challenge, initially reducing performance but ultimately enhancing long-term memory and skill transfer. Research has shown that specific learning strategies—such as retrieval practice (Roediger & Karpicke, 2006), spacing effects (Carpenter et al., 2012), and interleaved practice (Rohrer et al., 2020)—enhance learning by leveraging basic memory processes. In mathematics, interleaved practice is particularly effective in promoting cognitive flexibility and improving long-term memory (Mielicki & Wiley, 2022). This instructional strategy requires students to distinguish between problem types and select appropriate problem-solving strategies. A key mechanism behind desirable difficulty involves perceptual and conceptual disfluency, which can encourage deeper cognitive engagement. Previous research has shown that perceptual cues, such as spacing in arithmetic expressions, can implicitly reinforce mathematical hierarchical structure (Landy & Goldstone, 2007). Although congruent cues facilitate problem-solving fluency, incongruent cues may introduce desirable difficulty by requiring students to look beyond perceptual features to correctly apply mathematical principles. For example, students demonstrate less accurate problem solving when spacing is incongruent, rather than congruent, to the order of operations (Harrison et al., 2020), but it is unknown how their performance may be affected after the spacing cues are removed. Given that spacing and color may influence problem solving (Closser et al., 2022; Landy & Goldstone, 2007), the present study systematically examined their roles in mathematical learning, specifically in the context of the order of operations, to determine whether perceptual-conceptual incongruence can serve as a desirable difficulty to enhance learning.

### 1.1.3. Student Characteristics

Beyond perceptual cues and perceptual-conceptual congruence, individual student characteristics may also influence mathematical problem-solving. One key factor is *prior knowledge*, which plays a crucial role in how students benefit from instructional techniques, particularly worked examples (Barbieri et al., 2023). According to the expertise reversal effect (Kalyuga et al., 2003), instructional approaches that are effective for beginners may be less beneficial for advanced learners. Another important factor is the *perception of equivalence*, the ability to recognize mathematical structures through subtle visual cues. This skill has been linked to perceptual processing in mathematical reasoning (Goldstone et al., 2017; Kellman et al., 2010). Prior research suggests that individuals with stronger object-based attention deployment more accurately identify algebraic equivalence (Marghetis et al., 2016). Finally, *math anxiety* is a well-documented predictor of lower math performance, largely due to its impact on working memory capacity, particularly in tasks that require multiple operations (Ashcraft & Kirk, 2001; Hembree, 1990).

## 1.2. AIMS OF THE PROJECT

This research project investigated the individual and combined effects of two perceptual cues—spacing and color—under varying conditions of congruence with the order of operations. In our previous work, we examined whether and how congruent perceptual cues (i.e., spacing and color) affected students' performance on order-of-operations problems. We previously found that students with a higher perception of equivalence benefited more from congruent spacing cues than those with a lower perception of equivalence (Hornburg et al., 2025). Extending our prior research on the benefits of perceptual cues on mathematics performance, we integrated the perceptual cues into worked examples to examine their influence on students' immediate problem-solving performance, short-term learning, and long-term retention on order-of-operation problems (e.g., $4 - 7 \times 2 + 9$). The overall goals of the project were to:

1. Explore whether incongruent perceptual cues could provide desirable difficulties leading to improved learning and retention on order-of-operations problems
2. Identify whether and how combining multiple perceptual cues affects problem-solving performance, learning, and retention on order-of-operations problems within online contexts
3. Examine whether the effects of these perceptual cues on performance, learning, and retention on order-of-operations problems varied based on individual student characteristics

Using a 3×3 factorial design, we tested the effects of perceptual cues (spacing, color, or no cues) and cue congruence modes (congruent, neutral, or incongruent with the order of operations) while also exploring potential moderation effects of individual student characteristics. Based on prior literature, we predicted that (1) all students would demonstrate learning gains from pretest to posttest after the intervention, (2) congruent cues would support students' intervention performance and short-term learning, whereas incongruent cues would present challenges that promote deeper understanding, leading to greater long-term retention, and (3) student characteristics would interact with perceptual cue conditions to moderate the effects on students' performance, learning, and retention.

To address these study aims and hypotheses, the data reported here were collected through a randomized controlled trial (RCT) conducted in 2024 in a suburban school district in the southeastern U.S. The datasets contain information about 1,100 6th-grade school students, including their demographics, pre- and post-assessments, and performance during the intervention. Note that the full study description and findings are reported in another paper (Closser et al., under review). All intervention data are fully de-identified and stored on a project page on Open Science Framework (OSF) at https://osf.io/d3tzq. Additional data that include student demographics and assessments are available through a data sharing agreement (DSA).

## (2) METHODS

### 2.1 STUDY DESIGN

Students participated in four 30-minute sessions (see Table 1) over 9.63 weeks on average ($SD = 1.42$ weeks, *Max* = 11.90 weeks, *Min* = 6.43 weeks). Each session occurred approximately 1–2 weeks after the previous session. To accommodate varying classroom needs, some flexibility in this timeframe was permitted, resulting in small differences in the duration between sessions (Session 1-Session 2 (2.49 ± 0.94 weeks), Session 2-Session 3 (4.41 ± 1.30 weeks), and Session 3-Session 4 (2.73 ± 1.15 weeks).

| SESSION | STUDY PHASE | SUMMARY OF TASKS |
|---|---|---|
| Session 1 | Pretest | • Order-of-operations assessment (10 items)<br>• Postdiction of performance (2 items)<br>• Perceptual processing skills (12 items)<br>• Math anxiety (9 items)<br>• Math value (6 items) |
| Session 2 | Intervention | • In condition: Practice problems with worked examples (21 items; one warm-up and 20 intervention problems) |
| Session 3 | Intervention | • In condition: Practice problems with worked examples (9 items); helpfulness rating of worked examples |
| | Immediate Posttest | • Order-of-operations assessment (10 items)<br>• Postdiction of performance (2 items) |
| Session 4 | Delayed Posttest | • Order-of-operations assessment (10 items)<br>• Postdiction of performance (2 items)<br>• Perceptual processing skills (12 items) |

**Table 1** Overview of study procedure.

Ottmar et al. *Journal of Open Psychology Data* DOI: 10.5334/jopd.139

During Session 1, participants' baseline knowledge of the order of operations was assessed, as well as student-level characteristics, including postdiction of performance, perceptual processing skills, and math attitudes (math anxiety and math value).

In Session 2, students were randomly assigned at the student level to one of nine experimental conditions based on the 3x3 research design that varied the combinations of color and spacing-based cues (congruent, neutral, or incongruent for each) (see Table 2) and completed an experimental math intervention. As shown in Table 2, problems with congruent color cues used an orange font to highlight the higher-order operator (multiplication or division) and its operands, aligning with the order of operations. Neutral spacing cues used uniform spacing throughout. Congruent spacing cues removed spacing between the higher-order operator and its operands, while maintaining uniform spacing elsewhere to align with the order of operations. In contrast, incongruent spacing cues removed spacing between the lower-order operators (addition and subtraction), while keeping uniform spacing elsewhere. During the intervention, participants solved 21 order-of-operations problems (one warm-up and 20 intervention problems) in their assigned condition. After answering each problem, participants received feedback on their accuracy and viewed a correct worked example of the problem.

Session 3 consisted of nine problems completed in the same experimental intervention format as Session 2, followed by an immediate posttest consisting of ten order-of-operations problems with no perceptual cues or correctness feedback provided, as well as two postdiction of performance items.

In Session 4, participants completed a delayed posttest, consisting of 10 order-of-operations problems, two postdiction of performance items, and 12 perceptual processing items. The 10 order-of-operations problems completed in the immediate and delayed posttests were identical to those completed during the pretest. Details for all intervention tasks, conditions, and assessments can be found in the Materials/Survey Instruments section.

## 2.2 TIME OF DATA COLLECTION
Data were collected between August 2024 and November 2024.

## 2.3 LOCATION OF DATA COLLECTION
Data were collected from one large suburban school district in the southeastern U.S.

## 2.4 SAMPLING, SAMPLE, AND DATA COLLECTION
The student-level dataset contains 1,110 sixth graders (typically 11–12 years old) from 16 teachers in nine middle schools, within one school district. Our team had an established relationship and memorandum of understanding set up with the specific school district. We recruited sixth-grade teachers via an email sent from the middle school math specialist including a flier describing the study and a link to an interest form for teachers to complete. All schools with at least one teacher interested in participating were included in recruitment. Students in the math classrooms of participating teachers were included in the study. Parents had the opportunity to opt their child out of the study and also participating students had the opportunity to not assent for our team to use their data in this research study.

Sixth-grade students in classes of participating teachers participated in this experimental study online during their math class time. Participants accessed study tasks on their school-provided devices via links provided to them by their teachers. Teachers who opted to participate in the study received email instructions and the links before each session on how to assist students with the session assignment. The school district provided demographic information for the 1,011 participating students. Table 3 shows the demographic information for the students.

## 2.5 MATERIALS/SURVEY INSTRUMENTS
All intervention materials and assessments were built using custom jsPsych plugins (De Leeuw et al., 2023) and hosted on Pavlovia (https://pavlovia.org), which supports online experiments and was used to provide intervention materials to participants remotely. Sample study materials and items from the research study can be found in the materials folder on OSF at https://osf.io/d3tzq.

### 2.5.1. Assessment materials
**Order-of-Operations Problem-Solving Assessment (Sessions 1, 3, and 4).** Students completed the same

| | | COLOR CUES | | |
|---|---|---|---|---|
| | | **CONGRUENT (CC)** | **NEUTRAL (NC)** | **INCONGRUENT (IC)** |
| **Spacing Cues** | **Congruent (CS)** | $10 + \mathbf{4 \times 8} - 3$ | $10 + 4 \times 8 - 3$ | $\mathbf{10 + 4 \times 8} - 3$ |
| | **Neutral (NS)** | $10 + \mathbf{4 \times 8} - 3$ | $10 + 4 \times 8 - 3$ | $\mathbf{10 + 4} \times 8 - 3$ |
| | **Incongruent (IS)** | $10 + \mathbf{4 \times 8} - 3$ | $10 + 4 \times 8 - 3$ | $\mathbf{10 + 4} \times 8 - 3$ |

**Table 2** Example problem as presented in each of the nine experimental conditions.

Note: Bolded numbers and symbols were shown in orange font. See Figure 1 for an example.

| | ALL | CSCC | CSIC | CSNC | ISCC | ISIC | ISNC | NSCC | NSIC | NSNC |
|---|---|---|---|---|---|---|---|---|---|---|
| | *n* = 1,110 | *n* = 151 | *n* = 127 | *n* = 92 | *n* = 112 | *n* = 117 | *n* = 122 | *n* = 128 | *n* = 139 | *n* = 122 |
| | (100.00%) | (13.61%) | (11.44%) | (8.28%) | (10.09%) | (10.54%) | (10.99%) | (11.54%) | (12.52%) | (10.99%) |
| **Gender** | | | | | | | | | | |
| Male | 489 | 63 | 64 | 41 | 60 | 48 | 44 | 58 | 53 | 58 |
| | (44.05%) | (5.68%) | (5.77%) | (3.69%) | (5.41%) | (4.32%) | (3.96%) | (5.23%) | (4.77%) | (5.23%) |
| Female | 522 | 77 | 54 | 45 | 43 | 56 | 69 | 59 | 65 | 54 |
| | (47.03%) | (6.94%) | (4.86%) | (4.05%) | (3.87%) | (5.05%) | (6.22%) | (5.32%) | (5.86%) | (4.86%) |
| Not reported | 99 | 11 | 9 | 6 | 9 | 13 | 9 | 11 | 21 | 10 |
| | (8.92%) | (0.99%) | (0.81%) | (0.54%) | (0.81%) | (1.17%) | (0.81%) | (0.99%) | (1.89%) | (0.90%) |
| **Race/Ethnicity** | | | | | | | | | | |
| Asian | 445 | 65 | 49 | 40 | 45 | 52 | 55 | 36 | 49 | 54 |
| | (40.09%) | (5.86%) | (4.41%) | (3.60%) | (4.05%) | (4.68%) | (4.95%) | (3.24%) | (4.41%) | (4.86%) |
| White | 314 | 34 | 37 | 24 | 31 | 35 | 35 | 47 | 37 | 34 |
| | (28.29%) | (3.06%) | (3.33%) | (2.16%) | (2.79%) | (3.15%) | (3.15%) | (4.23%) | (3.33%) | (3.06%) |
| Hispanic Latino | 169 | 24 | 17 | 20 | 16 | 12 | 18 | 23 | 22 | 17 |
| | (15.23%) | (2.16%) | (1.53%) | (1.80%) | (1.44%) | (1.08%) | (1.62%) | (2.07%) | (1.98%) | (1.53%) |
| Black | 43 | 10 | 5 | 2 | 7 | 3 | 4 | 5 | 5 | 2 |
| | (3.87%) | (0.90%) | (0.45%) | (0.18%) | (0.63%) | (0.27%) | (0.36%) | (0.45%) | (0.45%) | (0.18%) |
| Multiple Races | 35 | 6 | 10 | 0 | 4 | 0 | 1 | 6 | 4 | 4 |
| | (3.15%) | (0.54%) | (0.90%) | (0.00%) | (0.36%) | (0.00%) | (0.09%) | (0.54%) | (0.36%) | (0.36%) |
| American Indian | 5 | 1 | 0 | 0 | 0 | 2 | 0 | 0 | 1 | 1 |
| | (0.45% | 0.09% | (0.00%) | (0.00%) | (0.00%) | (0.18%) | (0.00%) | (0.00%) | (0.09%) | (0.09%) |

(Contd.)

| | ALL | CSCC | CSIC | CSNC | ISCC | ISIC | ISNC | NSCC | NSIC | NSNC |
|---|---|---|---|---|---|---|---|---|---|---|
| | n = 1,110 | n = 151 | n = 127 | n = 92 | n = 112 | n = 117 | n = 122 | n = 128 | n = 139 | n = 122 |
| | (100.00%) | (13.61%) | (11.44%) | (8.28%) | (10.09%) | (10.54%) | (10.99%) | (11.54%) | (12.52%) | (10.99%) |
| Not reported | 99 | 11 | 9 | 6 | 9 | 13 | 9 | 11 | 21 | 10 |
| | (8.92%) | (0.99%) | (0.81%) | (0.54%) | (0.81%) | (1.17%) | (0.81%) | (0.99%) | (1.89%) | (0.90%) |
| ESOL | 92 | 13 | 10 | 7 | 4 | 8 | 7 | 16 | 14 | 13 |
| | (8.29%) | (1.17%) | (0.90%) | (0.63%) | (0.36%) | (0.72%) | (0.63%) | (1.44%) | (1.26%) | (1.17%) |
| Gifted | 264 | 36 | 28 | 24 | 29 | 32 | 30 | 31 | 27 | 27 |
| | (23.78%) | (3.24%) | (2.52%) | (2.16%) | (2.61%) | (2.88%) | (2.70%) | (2.79%) | (2.43%) | (2.43%) |
| IEP | 92 | 9 | 15 | 10 | 10 | 6 | 4 | 13 | 11 | 14 |
| | (8.29%) | (0.81%) | (1.35%) | (0.90%) | (0.90%) | (0.54%) | (0.36%) | (1.17%) | (0.99%) | (1.26%) |
| IST | 72 | 11 | 7 | 4 | 10 | 6 | 9 | 8 | 10 | 7 |
| | (6.49%) | (0.99%) | (0.63%) | (0.36%) | (0.90%) | (0.54%) | (0.81%) | (0.72%) | (0.90%) | (0.63%) |

**Table 3** Student Demographic Information by Condition (N = 1,110).

*Note:* CSCC = congruent spacing congruent color, CSIC = congruent spacing incongruent color, CSNC = congruent spacing neutral color, ISCC = incongruent spacing congruent color, ISIC = incongruent spacing incongruent color, ISNC = incongruent spacing neutral color, NSCC = neutral spacing congruent color, NSIC = neutral spacing incongruent color, NSNC = neutral spacing neutral color, ESOL = English to Speakers of Other Languages, referring to the students who learned English in a non-English-speaking country or learned English as a second language in an English-speaking country, Gifted = Determined by the school district based on a nationally normed test that includes measures of mental ability, achievement, motivation, and creativity, IEP = Individualized Education Program status, referring to whether students are disabled or need special health care, IST = Instructional Support Team status, referring to whether students exhibit academic difficulties and need assistance from a team of teachers.

10 order-of-operations problems at pretest (Session 1), posttest (Session 3), and delayed posttest (Session 4). These problems were constructed as traditional order-of-operation problems with three operators and no perceptual cues (e.g., $10 + 4 \times 8 - 3$). Each problem included one addition and one subtraction operator. The third operator was either multiplication (9 problems) or division (1 problem). The problems that included multiplication counterbalanced the multiplication operation in the left (e.g., $11 \times 4 - 8 + 3$), middle (e.g., $4 - 7 \times 2 + 9$), and right (e.g., $9 - 3 + 7 \times 4$) positions. The problem with division as the third operator positioned division as the middle operation (e.g., $7 + 12 / 6 - 4$). Parentheses and exponents were not included in any problems in this study. Problems were presented in the same order for all students. Correct responses were scored as 1, and incorrect responses were scored as 0. Performance on pretest, posttest, and delayed posttest was calculated as the summed total of student accuracy, with 10 being the highest possible score and 0 the lowest. The inter-item reliability coefficients of the ten items were KR-20 = 0.81 at the pretest, 0.89 at the posttest, and 0.88 at the delayed posttest.

**Postdiction of Performance (Sessions 1, 3, and 4).** After completing the order-of-operation assessment, students completed two items related to the postdiction of their performance. First, they responded to the question: "You just answered 10 problems. How confident are you that you answered them correctly?" with a 4-point Likert-type scale: "Not confident at all", "A little bit confident", "Somewhat confident", "Completely confident" (adapted from Labuhn et al., 2010). They then responded to the question "How many of these 10 problems do you think you answered correctly?" with an integer of 0–10 (adapted from Bol et al., 2012).

**Perceptual Processing Skills (Sessions 1 and 4).** To assess perceptual processing skills, students completed the perceptual math equivalence task (PMET; 12 items). Perceptual equivalence refers to the ability to identify mathematical expressions as equivalent or non-equivalent as quickly as possible (Botelho et al., 2021; Bye et al., 2024; Chan et al., 2022). The PMET is a two-part task that presents algebraic problems where the physical spacing of symbols matches or mismatches the appropriate order of operations (Kirshner & Awtry, 2004; Landy & Goldstone, 2007). Each part includes six items. In Part 1, students were presented with two expressions and indicated whether those expressions were equivalent or not equivalent. In Part 2, students were shown a target expression and six alternative expressions. They were required to identify the alternative expression that was equivalent (three items) or not equivalent (three items) to the target expression. Correct responses were scored as 1, and incorrect responses were scored as 0. Performance on the PMET was calculated as the summed accuracy of all items, with 12 being the highest possible score. The inter-item reliability coefficients of the 12 items were KR-20 = 0.46 for the PMET at pre-assessment and 0.48 for the PMET at post-assessment.

**Math Anxiety (Session 1).** The anxiety that students feel about mathematics was measured using an adaptation of the Math Anxiety Scale for Young Children – Revised (MASYC-R; Ganley & McGraw, 2016). This nine-item measure assesses three types of feelings – negative reactions to math (3 items; e.g., "When it is time for math, my head hurts"), confidence in math learning (3 items; e.g., "I like being called on in math class"), and worry about math (3 items; e.g., "I get worried when I don't understand something in math"). Students responded to each item on a 4-point Likert-type scale ("No," "Not really," "Kind of," "Yes"). Three items on confidence in math learning were reverse-coded for higher scores to indicate greater math anxiety. Math anxiety was measured as the participant's mean response across all nine items (Cronbach's alpha = 0.81).

**Math Value (Session 1).** The value that students perceive toward mathematics was measured using six items adapted from Wigfield and Eccles (2000). These items measured students' intrinsic value (2 items; e.g., "In general, I find working on math assignments interesting"), utility value (2 items; e.g., "Compared to most of my other activities, what I learn in math is useful"), and attainment value (2 items; e.g., "For me, being good in math is important"). Students responded to each item using a 4-point Likert-type scale ("No," "Not really," "Kind of," "Yes"). Overall, math value was measured as the participant's mean response across all six items (Cronbach's alpha = 0.79).

**Perceived Helpfulness of the Worked Examples (Session 3).** After each intervention math problem, students were provided with an example of the correct process for solving that problem. After completing all experimental intervention items in Session 3, students were asked to assess the "perceived helpfulness" of those materials. Specifically, they responded to the item: "How helpful were these worked examples?" using a 4-point scale: "Not helpful at all", "A little bit helpful", "Somewhat helpful", "Very helpful".

The exact items of all assessment materials, including the order-of-operations problem-solving assessment, perceptual processing skills, math anxiety, and math value, are listed in an assessment materials document on the OSF repository (DOI 10.17605/OSF.IO/D3TZQ, URL:https://osf.io/d3tzq/).

## 2.5.2. Experimental materials

**Practice Problems and Worked Examples (Sessions 2 and 3).** Students completed 21 experimental practice problems (including one warm-up problem) in Session 2 and nine problems in Session 3. Each problem was presented using the perceptual cues for a given participant's assigned condition (see Table 2). These

problems were constructed as order-of-operation problems with three operators. Each problem included one addition and one subtraction operator. The third operator was either multiplication (Session 2: 18 problems; Session 3: 6 problems) or division (Session 2: 3 problems; Session 3: 3 problems). The third operator (multiplication or division) was counterbalanced between the left, middle, and right positions. Parentheses and exponents were not included in any problems in this study.

Problems were presented one at a time via a screen progression of: Screen 1: the problem is presented and the student enters their response; Screen 2: student is shown their response alongside the correct response and asked to evaluate their answer ("How did you do?"; immediate correctness feedback); Screen 3: student is shown a step-by-step worked example of the problem. See Figure 1 for examples of these screen displays.

Students completed the first problem in Session 2 prior to any instruction on the order of operations. This problem was presented to students as a "warmup" problem with the instructions: "In this section, you will solve a series of math problems. Try your best to simplify each expression. Answers could be positive or negative numbers." After entering their response, students were shown the correct response and worked example screens as described above. Session 3 followed the same format but did not include a warmup problem.

## 2.6 QUALITY CONTROL

Our datasets meet data quality standards in terms of quantity, quality, and accuracy. They include a large amount of student data ($N$ = 1,110) collected through an RCT conducted in a large school district. To ensure data accuracy, students were required to enter only numeric responses to move on to the next screen during problem-solving tasks. To maintain data quality, our research team de-identified and pre-processed the dataset. We first handled the entry errors in students' IDs (PID) and teachers' names. For example, some students incorrectly completed the fields for their PIDs (e.g., 123456) or teachers' names (e.g., "I don't want to tell you"). Additionally, a student may input the PID correctly in sessions 1, 3, and 4 (e.g., 137579) but incorrectly in session 2 (e.g., mistype 137579 as 137570). To handle such cases ($n$ = 10), our research team manually 1) checked with the school district on whether 137570 is a valid PID, 2) confirmed that the PID 137570 and PID 137579 are associated with the same teacher, and 3) confirmed that except for 137570, no other similar PID (e.g., 137578) can compensate for the missing session associated with the PID 137579. Then, we replaced student IDs used in the school district with randomly generated IDs consisting of EHR followed by four-digit numbers (e.g., EHR0001). Quality control efforts also included making design decisions that were informed by lessons learned from a prior study (Hornburg et al., 2025), where our research team collected student performance and teacher feedback for a sample of similar order-of-operations problems. Changes made based on the prior study included changing the online platform used for the study and adjusting verbiage to improve the clarity of instructions.

## 2.7 DATA ANONYMISATION AND ETHICAL ISSUES

This study was approved by the Institutional Review Board (IRB) at each affiliated university, and the primary

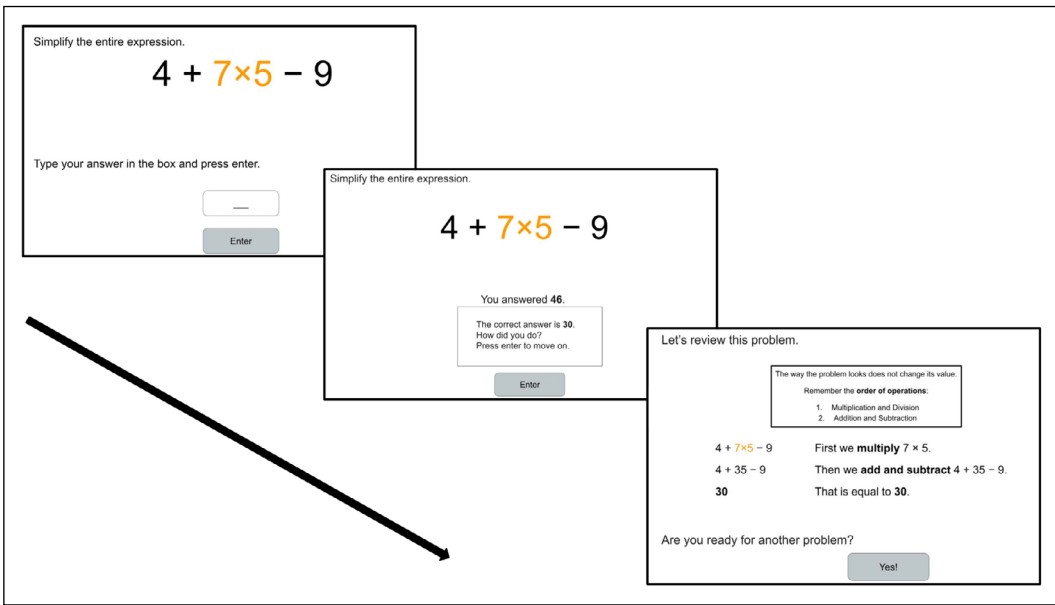

**Figure 1** Example Screen Progression from a Practice Problem to Immediate Correctness Feedback to Worked Example (Congruent Spacing, Congruent Color [CSCC] Condition).

(Closser et al., under review).

institution oversees the study under the "research conducted in established educational settings" exemption. Teachers provided informed consent to participate in the study. Prior to giving their informed consent, teachers were given an overview of the study requirements and time commitment. A letter was sent home to parents providing information about the study and the contact information for the district administrator, who was our primary contact for this project. The letter also informed parents that to opt their child out of participating, they needed only to email the district administrator. A study overview with screenshots of the types of questions their child would be responding to was provided to the district to share with parents who preferred additional information about the study before choosing whether to opt their child out. On the first day of the study, students were provided with a written description of the study activities and purpose, and they gave their assent to participate. Students who did not provide assent were given a non-research math activity by their teacher. At the beginning of the study, each student was assigned a de-identified participant identification number. Students' names were never recorded.

## 2.8 EXISTING USE OF DATA

Closser, A. H., Bye, J. K., Zhang, P., Lee, J., Egorova, A., Colbert, M. A., Yu, S., Hornburg, C. B., & Ottmar, E. (under review). *The effects of congruent and incongruent perceptual cues on middle schoolers' mathematical performance, learning, and retention.*

Hornburg, C. B., Zhang, P., Closser, A. H., Bye, J. K., Lee, J.-E., Egorova, A., Colbert, M. A., Yu, S., & Ottmar, E. (2025, June). Understanding the effects of perceptual cues on middle schoolers' mathematical performance, learning, and retention. In A. H. Closser (Chair), *Leveraging cognitive principles in low-cost interventions to improve mathematics problem solving and learning.* Paper to be presented at the 8th Annual Meeting of the Mathematical Cognition and Learning Society (MCLS), Hong Kong, China.

In these papers, we addressed the project aims outlined in Section 1.2 and tested the unique and combined effects of spacing and color cues across three modes of congruence with the order of operations, as well as the moderating effects of student characteristic variables. Specifically, we used student-level assessment data (i.e., pretest, perception of equivalence skills, posttest, delayed posttest, and math anxiety), performance data from the intervention, and demographic data to address our research questions. For the data analysis, we re-ran the models using the Neutral Spacing and Neutral Color (NSNC) condition as the reference group, comparing performance, learning, and retention across eight dummy-coded experimental groups to make the results easier to interpret and to enable clearer comparisons across conditions. See Closser et al. (under review) for more details on the data analyses and results.

# (3) DATASET DESCRIPTION AND ACCESS

This data provides information about the study at two levels: (1) problem-level data, which includes student actions, responses, and response time per item, and (2) student-level data, which includes aggregated student actions, response time, and demographics per participant. We have organized the data in a way that allows for multiple levels of analysis and the selection of variables of interest.

## 3.1 REPOSITORY LOCATION

Information and files for the study description, assessment materials, and data dictionary describing the data are located on the OSF website (DOI 10.17605/OSF.IO/D3TZQ, URL: https://osf.io/d3tzq/) and can be accessed on the project repository. A subset of the data, including the experimental intervention items, is also available on this OSF page. The full dataset that includes the students' assessments and demographic information requires the data sharing agreement and is stored on a second OSF repository and is available with a separate link that is emailed to the researcher once the DSA is received. For access to the full dataset, data sharing agreement forms and instructions are also available on this page. To access and download the full data files, including student demographic and assessment data, researchers must complete the Data Sharing Agreement (DSA) form and email their signed agreements to the project team (described in more detail in section 3.7 below). The additional step is required by the IRB in order to ensure that the de-identified data files can be available for sharing while meeting ethical requirements. Once the request process is completed, a link to the actual data files will be sent to the researcher by the project team.

## 3.2 OBJECT/FILE NAME

As described above, all intervention data are openly available on the OSF repository (https://osf.io/d3tzq/). This open dataset consists of two separate files: problem-level data from intervention problems in Session 2 (training_1.csv) and problem-level data from Session 3 (training_2.csv). Table 4 provides a description of each variable included in the open intervention dataset.

The second dataset that requires the DSA includes students' assessment and demographic information. It is organized into two student-level datasets and 11 problem-level datasets, totalling 13 separate files. The contents within each file are described in Table 5.

| VARIABLE | DESCRIPTION | TYPE |
|---|---|---|
| stu_id | Student ID assigned in the study (non-identifiable) | character |
| Session | The session of the task | character |
| task | The name of the task | character |
| stim_num | Problem number | character |
| response | Students' response for each problem or survey item | character |
| correct | Correctness of each problem | dichotomous |
| rt | Time taken to solve each problem (in milliseconds) | continuous |
| rt_fd | Time taken to view the feedback after each problem (in milliseconds) | continuous |
| rt_we | Time taken to view the worked example after each problem (in milliseconds) | continuous |
| condition | Assigned condition (e.g, CSCC, CSIC) | categorical |

**Table 4** Description of each variable in the available intervention dataset on OSF.

| FILE NAME | CONTENTS | STRUCTURE |
|---|---|---|
| **student-level dataset** | | |
| student_all | Student performance data on pre, intervention, post, and delayed posttest | one row per student (wide format) |
| student_demo | Data provided by the school district on student demographics (e.g., gender, race/ethnicity) | |
| **problem-level dataset** | | |
| problem_meta | Metadata on the intervention problems (e.g., problem order, problem type) | one row per problem |
| pretest | Problem-level data on students' prior knowledge of the order of operations | one row per student per problem (long format) |
| pre_pmet | Problem-level data on students' prior perception of equivalence skills | |
| pre_assess | Problem-level data on pre-math anxiety, pre-math value, postdiction of performance | |
| training_1 | Problem-level data on intervention problems (session 2) (publicly available on OSF) | |
| training_2 | Problem-level data on intervention problems (session 3) (publicly available on OSF) | |
| posttest | Problem-level data on students' posttest knowledge of the order of operations | |
| post_pmet | Problem-level data on students' post-perception of equivalence skills | |
| post_assess | Problem-level data on post-helpfulness rating and postdiction of performance | |
| delayed_post | Problem-level data on students' delayed posttest knowledge of the order of operations | |
| delayed_assess | Problem-level data on delayed posttest helpfulness rating and postdiction of performance | |

**Table 5** Description of each file in the full dataset that includes student assessment and demographic information (accessible once the DSA is completed).

The student-level datasets consist of two files: (1) data on student pretest, intervention, posttest, and delayed posttest ("student_all" in Table 5), and (2) data on their demographics ("student_demo" in Table 5). Student data (student_all) provide overall measures of how well each student performed on each task or how they self-reported their perceptions. Demographic data (student_demo) were provided by the school district and include information on students' biological sex, race/ethnicity, English as a Second Language (ESL) status, their 5th-grade state assessment math score, and other relevant variables. Their 5th-grade state assessment scores include overall numeric scores as well as seven performance levels that were provided by the district (i.e., below target, approaching target, met target) for subdomains of mathematics, including

numerical reasoning, pattern and algebraic reasoning, measurement and data reasoning, and geometric and spatial reasoning.

The problem-level dataset is structured by the data collection time points (e.g., pre, post) as well as the task type (e.g., knowledge of the order of operations, perception of equivalence skills, math anxiety). Hence, these data provide detailed information on how well each student performed on each *problem* and how they self-reported their perceptions of each *item*. Details about other measures included in each file are documented in the data descriptions on OSF (https://osf.io/d3tzq/).

### 3.3 DATA TYPE

These datasets contain primary data from the randomized controlled trial. They include various types of data, including students' demographics, assessment, and performance during the intervention, collected from multiple sources. The datasets comprise four types of data: raw clickstream data, pre-processed data, students' self-reported data, and data provided by the school district.

The problem-level data files are derived from the raw clickstream data extracted from the Pavlovia platform, where the study was hosted and conducted. Student data on pretest, intervention, posttest, and delayed posttest were pre-processed and aggregated from the problem-level log data. Assessment data on math anxiety, math value, helpfulness ratings, and postdiction of performance were self-reported by students on the platform. All demographic data were provided by the students' school district.

### 3.4 FORMAT NAMES AND VERSIONS

All 13 data files are in the comma-separated value (CSV) file format. Each file contains unique anonymous identifiers of students (e.g., EHR0001), sessions (e.g., S1, S2), or problems. Researchers can merge multiple layers of data using these identifiers for specific data analyses. For example, to investigate demographic differences in student performance on intervention problems, a researcher could merge the "student_all" and "student_demo" files using the student identifier (i.e., stu_ID). Alternatively, to examine how different types of problems affect student performance on intervention problems, they would merge the "problem_meta", "training_1", and "training_2" files using the student identifier as well as the problem identifier (i.e., stim_num).

### 3.5 LANGUAGE

The data are stored in American English.

### 3.6 LICENSE

The data descriptions, study description, a subset of intervention data, and request procedures and forms to access the full study data are available on OSF and are available under a CC0 license. However, due to restrictions imposed by our institution's IRB and the Memorandum of Understanding (MOU) with the participating district, the full data cannot be made completely open and are not under a CC0 license (i.e., no copyright reserved). However, access to the full deidentified data from this study is publicly available on OSF with a signed DSA. The details of the DSA process are described on the project OSF page, and the project Principal Investigator (PI) at the primary institution manages the approval process.

### 3.7 LIMITS TO SHARING

As mentioned above, a subset of the data that includes problem-level responses and condition assignment for each student is openly available for researchers to download on OSF (https://osf.io/d3tzq/). However, the full dataset containing assessment and demographic data is only available to researchers who have completed the DSA process with the PI's university. The DSA is required by the PI's institution's IRB as well as the signed MOU with the school district. As student-level data are protected under the Family Educational Rights and Privacy Act (FERPA) in the U.S., the PI's institution requires researchers to sign an agreement stating that they will not share the data with others.

To access the full dataset, researchers must complete a DSA form (i.e., data sharing agreement) available on our OSF page (https://osf.io/d3tzq/) and submit it to the project PI along with evidence of an ethics training certificate. One commonly used example of such a training is the Collaborative Institutional Training Initiative (CITI) training (Human Subjects in Social & Behavioral Research course) (https://about.citiprogram.org). CITI training on the protection of human subjects in research is required by university IRBs for most faculty, staff, and students in the United States before conducting any research, is available to anyone aged 18 and older, and takes approximately 2–3 hours to complete. After completion, CITI provides a certificate that can be downloaded and sent to the IRB as evidence of ethics training. For researchers in institutions or countries where the CITI program is unavailable or not required, any equivalent ethics certificate or evidence of human subjects training is accepted. Once the signed DSA forms and ethics certificates are submitted to the PI, the researchers will receive an emailed link to access the full de-identified datasets. The estimated processing time for the DSA is 2 to 3 business days. In the case that the PI changes institutions, the OSF link will be updated with the new contact information and forms.

### 3.8 PUBLICATION DATE

The open intervention datasets and study materials were added on 7/15/2025. Public DSA Request processes and forms for the full datasets were added on 5/22/2025. The

full DSA-approved dataset was also added to the second OSF page on 5/22/2025.

### 3.9 FAIR DATA/CODEBOOK
Our dataset meets FAIR data standards, ensuring it is findable, accessible, interoperable, and reusable (Wilkinson et al., 2016).

**Findability:** The data are easily findable on OSF and include unique identifiers of students, sessions, or problems, which allows researchers to merge multiple layers of data using these identifiers for specific data analyses. The content and description of each column in all data files are clearly documented in our data dictionary, which is available on a searchable webpage on OSF (https://osf.io/d3tzq).

**Accessibility:** The intervention data and study materials are easily accessible through a link on OSF, and the protocol for accessing the full data is free and universally open-sourced through the authorization of our university required DSA protocol. Metadata (i.e., data dictionary) will remain available on OSF even after the intervention study is no longer accessible due to MOU agreements and the termination of the IRB approval.

**Interoperable:** The datasets are saved in CSV format with multiple layers and can be effectively analyzed using various statistical or data analysis packages. Our metadata (i.e., data dictionary) provides column names, descriptions, data types (e.g., character, categorical, continuous), categorical values with corresponding codes (e.g., No = 0, Not really = 1, Kind of = 2, Yes = 3) as well as data sources, ensuring clarity and interpretability for other researchers.

**Reusable:** The dataset is readily reusable, given its large sample size, comprehensive measurements, and systematic procedures for the manipulation of perceptual cues. Please see the section below titled "Reuse potential" for additional details regarding reusability.

## (4) REUSE POTENTIAL

### 4.1 STRENGTHS OF THE DATASET
This dataset possesses great potential for reuse in educational research, given its large sample size, comprehensive measurements, and systematic manipulation of perceptual cues. First, this dataset includes over 1,000 middle school students with comprehensive measurements, such as demographic information provided by the school district, math problem-solving measures, as well as self-ratings of math anxiety and math value. Second, this dataset records students' performance during pretest, intervention with perceptual cues, posttest, and delayed posttest, which allows researchers to investigate the impact of perceptual cues on immediate problem-solving performance, math learning, and long-term retention. Moreover, two visual

attributes (i.e., color and spacing) of math expressions were manipulated, resulting in a 3×3 factorial design, i.e., three levels of color cue (congruent, neutral, incongruent) × three levels of spacing cue (congruent, neutral, incongruent). This 3×3 factorial design allows the investigation of how different combinations of perceptual cues impact problem-solving, learning, and retention.

### 4.2 SECONDARY ANALYSIS
This dataset can be used to conduct multiple secondary analyses focusing on math learning and perceptual cues. For example, by analyzing student-level data, researchers can explore how individual differences (e.g., 5th-grade numerical reasoning, pattern and algebra reasoning, measurement and data reasoning, geometric and spatial reasoning scores) influence the learning of operational rules under different perceptual cues. Additionally, as the position of the high-order operator in the order-of-operations problems is manipulated, the problem-level dataset can be used to examine how different locations of higher-order operators (e.g., left, middle, or right) impact problem-solving performance and how these interact with different perceptual cues (e.g., Closser et al., 2024). In terms of data analysis methods, further research can apply more sophisticated educational data mining techniques. For example, students' intervention performance can be used to train knowledge-tracing models to further understand how perceptual cues impact math learning (i.e., hidden knowledge states) and immediate problem-solving performance (i.e., guessing or slipping rate) as intervention proceeds. Finally, after the perceptual cue-involved intervention, students rated the helpfulness of worked examples. Researchers can also explore how perceptual cues and students' characteristics (e.g., prior knowledge, math anxiety, or in-class math performance) influence their rating scores on the helpfulness of worked examples. In sum, through these potential secondary analyses, this dataset can provide insight into the application of perceptual scaffolding in math education, as well as the way of tailoring perceptual scaffolding based on individual characteristics.

### 4.3 THEORETICAL IMPLICATIONS
The current dataset can promote the understanding of the perceptual learning theory in math (Closser et al., 2022; Goldstone et al., 2017; Kellman et al., 2010). Previous literature has commonly manipulated spatial (Braithwaite, 2016; Jiang et al., 2014; Landy et al., 2008; Landy & Goldstone, 2010) or color proximity (Alibali et al., 2018) of mathematical terms to create perceptual cues during math problem-solving or math intervention. Our dataset includes the manipulation of color and spacing cues, which are two different visual attributes and may influence math reasoning

differently. For example, unbalanced spatial proximity can spontaneously appear in the handwriting of math expressions (Landy & Goldstone, 2007), while the color of mathematical terms within one math expression is relatively less likely to differ. Moreover, spatial processing and math processing are highly overlapped (for a review, see Hawes & Ansari, 2020). Thus, the effect of spacing cues may reflect students' perceptual *learning* driven by daily math problem-solving experience apart from bottom-up perceptual grouping, while the effect of color cues may be relatively limited to the perceptual *grouping* mechanism. Studying the difference between spatial and color cues, as well as the individual differences in the reliance on spatial and color cues, illuminates the associations between low-level perceptual pathways and high-level math reasoning.

### 4.4 POTENTIAL COLLABORATION AND VALIDATION

Our datasets can serve as a means for new collaborations between researchers and the project team for replication research or follow-up research. For example, future studies focusing on perceptual cues can refer to this study. Since the perceptual cue condition was manipulated between subjects, researchers can select subgroups of students from certain cue conditions (e.g., CSNC versus NSNC versus ISNC) to examine the effect size of one type of perceptual cue (e.g., spacing). In addition, this dataset can serve as a control condition for future studies that aim to improve perceptual cue-based intervention, such as providing personalized perceptual cues adapted to students' learning progress. Moreover, collaborators who are interested in designing theory-based educational tools can use this dataset to understand the effects of different perceptual scaffolding (e.g., congruent spacing or congruent color) on problem-solving and math learning.

### 4.5 LIMITATIONS

Despite containing rich assessments and a large sample size, our dataset still exhibits several limitations. One major limitation is the presence of missing data. Of the 1,110 students, only 537 (48.38%) of them completed all assessments across four sessions without any missing data points. The low completion rate may be partly due to the extended data collection period, which spanned on average of 9.62 weeks ($SD$ = 1.43 weeks). Another critical issue resulting in missing values is the entry errors in students' PIDs and teachers' names. Since the PID was used as the identifier of each student, an error in the PID led to the failure of matching students across sessions. A second drawback concerns the low inter-item reliability of the perceptual processing skills assessment (i.e., PMET), which may be due to the limited number ($n$ = 12) of items included in the test or their varied formats. Additionally, we employed the same set of order-of-

operations problems in the pretest, posttest, and delayed posttest, which may potentially introduce the memory effect on the posttest performance.

## ACKNOWLEDGEMENTS

We thank the teachers, students, and district for their participation in this project.

## FUNDING STATEMENT

The research reported here was supported by the National Science Foundation through a grant, "Examining the Effects of Perceptual Cues on Middle School Students' Online Mathematical Reasoning and Learning" (#2300764) to Worcester Polytechnic Institute. The opinions expressed are those of the authors and do not represent the views of the National Science Foundation.

## COMPETING INTERESTS

The authors have no competing interests to declare.

## AUTHOR CONTRIBUTIONS

Erin Ottmar: Funding acquisition, Project administration, Supervision, Writing original draft, Review, and Editing.
Puyuan Zhang: Data curation, Writing original draft, Review, and Editing.
Ji-Eun Lee: Funding acquisition, Project administration, Supervision, Data curation, Writing original draft, Review, and Editing.
Jeffrey K. Bye: Funding acquisition, Methodology, Data curation, Software, Review and Editing.
Maegan A. Colbert: Writing original draft, Review, and Editing.
Alena Egorova: Review and Editing.
Shuqi Yu: Writing original draft, Review and Editing.
Avery H. Closser: Funding acquisition, Project administration, Writing original draft, Review, and Editing.
Caroline Byrd Hornburg: Funding acquisition, Project administration, Supervision, Writing original draft, Review, and Editing.

## AUTHOR AFFILIATIONS

**Erin Ottmar** ⓘD orcid.org/0000-0002-9487-7967
Department of Social Science and Policy Studies, Worcester Polytechnic Institute, US
**Puyuan Zhang** ⓘD orcid.org/0000-0003-3217-2870
Department of Social Science and Policy Studies, Worcester Polytechnic Institute, US

**Ji-Eun Lee** [ID] orcid.org/0000-0001-8521-8997
Department of Social Science and Policy Studies, Worcester Polytechnic Institute, US

**Jeffrey K. Bye** [ID] orcid.org/0000-0002-2636-3657
Department of Psychology, California State University, Dominguez Hills, US

**Maegan A. Colbert** [ID] orcid.org/0009-0003-0456-3767
Department of Human Development and Family Science, Virginia Tech, US

**Alena Egorova** [ID] orcid.org/0000-0002-4577-9615
Department of Social Science and Policy Studies, Worcester Polytechnic Institute, US

**Shuqi Yu** [ID] orcid.org/0000-0001-7469-8876
Department of Human Development and Family Science, Virginia Tech, US

**Avery H. Closser** [ID] orcid.org/0000-0003-2712-8509
School of Teaching and Learning, University of Florida, US

**Caroline Byrd Hornburg** [ID] orcid.org/0000-0001-9563-859X
Department of Human Development and Family Science, Virginia Tech, US

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

## PEER REVIEW COMMENTS

*Journal of Open Psychology Data* has blind peer review, which is unblinded upon article acceptance. The editorial history of this article can be downloaded here:

- **PR File 1.** Peer Review History. DOI: https://doi.org/10.5334/jopd.139.pr1

Ottmar et al. *Journal of Open Psychology Data* DOI: 10.5334/jopd.139

**TO CITE THIS ARTICLE:**
Ottmar, E., Zhang, P., Lee, J.-E., Bye, J. K., Colbert, M. A., Egorova, A., Yu, S., Closser, A. H., & Hornburg, C. B. (2025). Data from the Effects of Congruent and Incongruent Perceptual Cues on Middle Schoolers' Mathematical Performance, Learning, and Retention Study. *Journal of Open Psychology Data,* 13: 5, pp. 1–16. DOI: https://doi.org/10.5334/jopd.139

**Submitted:** 28 May 2025    **Accepted:** 01 August 2025    **Published:** 13 August 2025

