## [Peer Review History. · Journal of Open Psychology Data]

Peer Review Comments for "Data from the Examining the Effects of Perceptual Cues on Middle School Students' Online Mathematical Reasoning and Learning Study"

Dear Erin Ottmar, Puyuan Zhang, Ji-Eun Lee, Jeffrey Bye, Maegan Colbert, Alena Egorova, Shuqi Yu, Avery Closser, Caroline Byrd Hornburg,

After review, we have reached a decision regarding your submission to Journal of Open Psychology Data, "Data from the Examining the Effects of Perceptual Cues on Middle School Students' Online Mathematical Reasoning and Learning Study ". Our decision is to request revisions of the manuscript prior to further consideration for publication.

The full review information is included at the bottom of this email. Please note that there may also be a copy of the manuscript file with reviewer comments available once you have accessed the submission account. In principle, the reviewers raised concerns about the availability of the data and provided a number of potential suggestions to manage this concern. They also provided some helpful comments that will help refine the clarity of the manuscript. As such, I ask you to consider all their comments and come to a clear action on the data accessibility before resubmitting. I will try to avoid back and forth decision-making so I will take your next submission to be the best attempt possible to conform to the journal requirements and make a final decision from this.

Instructions for how to resubmit your article online are pasted below. Please ensure that your revised files adhere to our author guidelines, and that the files are fully proofed prior to upload. Please also include a revised version of your article with 'tracked changes', adding comments where appropriate, to indicate the revisions made, in addition to a brief document outlining how you have responded to the reviewers' requests.

If you have trouble processing the revisions, our Help Center (<https://help.u-community.io>) or downloadable PDF (<https://bit.ly/Author-Guide-OJS-3>) may be able to help. If not, please get in touch and we'll be happy to help.

Please also ensure that all copyright permissions have been attained for any figures/tables you have included.

Please could you have the revisions submitted with six weeks. If you cannot make this deadline, please let us know as early as possible.

Kind regards,

Prof Thomas Rhys Evans

R1:

This paper presents the datasets of a randomized control trial to improve 6th-graders' understanding of "order-of-operation" problems. The datasets are described as reporting detailed demographics and math-related individual characteristics and pre-test, intervention, and post-test performance. These datasets are interesting and have several potential reuses that can be useful for researchers in the field.

However, it is crucial to note that the presented datasets are not directly accessible in OSF. To obtain access to the datasets, users must complete two forms and provide a certificate. The datasets will be made available within 3 workdays after these documents are provided. This procedure is imposed by authors' institution.

I think that this procedure does not satisfy two points of the journal guidelines:

- The data must be deposited under an open license that permits unrestricted access (e.g. CC0, CC-BY).
- The deposited data must include a version that is in an open, non-proprietary format.

The fact that one has to send documents and does not have immediate access to the dataset is not consistent with "unrestricted access" (the authors themselves state that "the full datasets are available on an OSF page but are not under a CC0 license") and "open" format. I find that this procedure is against the "Accessibility" FAIR principle as it can be too complicated for potential users. For example, one should understand the certificate needed and how to get it from one's own institution, and bureaucracy may be difficult to handle.

Can the authors make a subset of the datasets openly available and keep this request procedure only for the full datasets? This could be a way to adhere to the journal guidelines, if the authors' institution allows this.

This is the reason why I recommend to resubmit this study. According to the description provided, the datasets look rich and valuable and they deserve to be made easily accessible.

Moreover, I suggest the authors to work on the manuscript, to enrich the background and provide a more exhaustive description of the variables in the dataset. You can find here a list of comments and suggestions.

1) Background

- You mention "Extending our teams prior research on the benefits of perceptual cues on mathematics performance". Please briefly describe this research and the results.
- "problem-solving performance". What kind of problem solving?
- "short-term learning, and long-term retention". What is supposed to be learned and retained?
- Move paragraph 1.1 after 1.2, because without background it is difficult to understand the aims.
- Provide a definition of "order-of-operation problems"
- I suggest to move the hypothesis with the study aims. A possibility is to describe the students' characteristics you measured in paragraph 1.2.3, as you have already done, but deleting the hypothesis from there. In the "Aims" section, you could add a paragraph starting like "In

addition, this data allow to investigate students characteristics, such as We hypothesise that". I hope this helps.

2) Study design

- How were students or teachers recruited? Please describe how you got in touch with schools or teachers

- Was condition assignment at the level of the teacher/class or of the students?

- Table 2. I am not sure I understand the difference between the three spacing cues. Did you manipulate the spacing between single numbers and operation signs? Adding a note below the table explaining the difference between the different cues, both for spacing and color, can be helpful.

- Table 3. Instead of "CS = congruent spacing, IS = incongruent spacing, NS = neutral spacing, CC = congruent color, IC = incongruent color, NC = neutral color", I think it may be more straightforward to explain the complete acronyms of the groups (CSCC, CSIC...), so that it is more directly related to the text in the table. I understand you wanted to be more syntetic, but it is not straightforward as it is now, because in the table one does not read for example "CS" or "NC" separately.

- There are some measures in the dataset that are not described here, for example numerical reasoning and geometric and spatial reasoning. Please provide a description of all materials used for all the variables reported in the dataset

- "Changes made based on the prior study included changing the online platform used for the study and adjusting verbiage to improve the clarity of instructions.". Why did you change the online platform?

- Did you somehow check whether the students were engaged in the tasks, whether they put enough effort?

3) Dataset description and access

- In "Object/file name" also list the files you share in OSF and to which data file they refer to

4) Potential reuse

- You describe how you dealt with the problem of students providing incorrect ID. This is an information to report in the Study design, maybe in "Data quality". Here you should mention limitations that your dataset currently has. For example, a limitation is the presence of missing data because of this problem. Or the fact that you don't have a delayed follow-up to test whether improvements hold in time. Moreover, I suggest to place this paragraph at the end of section 4.

R2:

In this manuscript, the authors are describing a large dataset of school children solving mathematical problems. The dataset is between-subjects and longitudinal, allowing to investigate how spacing and colour cues under varying conditions (congruent, neutral, or incongruent) affect students' performance.

Overall, I found the manuscript clearly written. I could follow the description of the methodology, with only small suggestions for improvement (listed below).

However, when I reached the description of the dataset, I encountered a major issue. The Journal of Open Psychology Data is aiming to published openly accessible datasets that comply with the FAIR principles. I do not think the current manuscript fulfils those principles, in particular the principle of accessibility. While there is an OSF project associated with the current manuscript, it contains only the material and data descriptions. There are no actual data there. To access the data, interested researchers need to i) complete two DSA forms (data sharing request, data sharing agreement) and ii) submit their Collaborative Institutional Training Initiative (CITI) training certificate (Human Subjects in Social & Behavioral Research course). Both the DSA forms and the CITI certificate need to be sent to the PI of the project by email, who then examines them and decides whether the access should be granted or not.

There are multiple problems with these requirements and the accessibility statement. First, the authors state that their data are "easily accessible" which is misleading as it requires the DSA forms and the CITI certificate. Completing the forms could be understandable if the dataset contained sensitive data. Since the authors claim their data have been anonymised and students are only identified with a randomly generated identifier, I do not see how this constitutes sensitive data. The CITI certificate is the biggest issue here. As the authors state, it is only accessible for the US based scientists (and it might be paid, although I could not find this information on the CITI website, which appeared very commercial). According to the authors, scientists from other countries can submit an equivalent certificate. The problem is, such certificates might not exist in other countries, or at least, I am not aware of them in Europe. The CITI training itself is 2-3 hours long, posing a significant barrier to the data accessibility.

Second, these submitted forms need to be treated by the PI. Although the authors state that the request will be treated within 2-3 working days, can this truly be guaranteed? I know how busy the academic life gets, and there are many urgencies and priorities each day. Such request emails risk getting lost with hundreds of other emails. I also understand that the PI retains the right to reject the request, although the authors do not comment on this aspect. Under which conditions would requests be granted or rejected (e.g., does it depend on the country of origin of the asking researcher)? I am also worried about what happens when the PI decides to change institutions or retires. It does not seem that the data will be accessible long term. In fact, the authors comment on this point in the accessibility section by acknowledging that the data will become inaccessible when "IRB approval terminates" and "MOU agreements" end. This directly violates FAIR principles, which emphasize persistent accessibility.

Overall, the authors appear to misunderstand FAIR principles by conflating "having a process" with "being accessible." True FAIR accessibility would require either open access or justified minimal restrictions with long-term preservation guarantees. The current dataset is only temporarily available to some researchers, and only as long as they fulfil the conditions

(conditional access). In the future, only the metadata will remain accessible, which is insufficient for any future reuse of the data.

I do understand that these requirements are likely imposed by the local ethics committee. Having such requirements is probably fine for articles that report on project findings. However, as the main point of a data paper is to describe an openly accessible dataset, I do not think this work has much value under such restrictive and temporary conditions.

Other minor suggestions:

- In the background section, I would move the aims of the project part towards the end, likely before the student characteristics section. For naïve readers, the aims of the project section contains a lot of undefined terms, which get explained a bit later, and thus, the section would make more sense after having read that information.
- In the method section (study design), I would clarify the age of participants. I know the authors state they are “sixth grade students” but grade system highly depends on the country, so to make it clear for all the readers, I would add the age range. In the same section, I would clarify that they ran 3x3 between-subjects experimental design.
- In the location of the data collection section, it would be helpful to have slightly more precise information where the schools were located. Could the authors at least give the state or even a town/city, if they are not allowed to name the actual school (which of course would give the most precise information)?
- In Table 3, I would add the definition for “Gifted” students. It might be a US-specific term, but I am not familiar with it, so knowing how giftedness is assessed would be helpful.
- In the data anonymisation and ethical issues section, I wonder if the authors meant to write “consent” instead of “assent”.
- In the existing use of data section, I would have liked to have a brief description of which data were used in these papers in preparation and which research questions are being answered.
- In the language section, I believe it should say “data are stored” instead of “data is stored” since the word data is a plural form of datum.